# Assessing the Socio-Economic Benefits and Costs of Insect Meal as a Fishmeal Substitute in Livestock and Aquaculture

**DOI:** 10.3390/ani14101461

**Published:** 2024-05-14

**Authors:** Alberts Auzins, Ieva Leimane, Rihard Reissaar, Jostein Brobakk, Ieva Sakelaite, Mikelis Grivins, Lauma Zihare

**Affiliations:** 1Institute of Agricultural Resources and Economics, LV-1039 Riga, Latvia; ieva.leimane@arei.lv (I.L.); lauma.zihare@arei.lv (L.Z.); 2Institute of Agricultural and Environmental Sciences, Estonian University of Life Sciences, 51006 Tartu, Estonia; rihard.reissaar@emu.ee; 3Ruralis—Institute for Rural and Regional Research, 7049 Trondheim, Norway; jostein.brobakk@ruralis.no; 4Institute of International Relations and Political Science, Vilnius University, LT-01130 Vilnius, Lithuania; ieva.sakelaite@tspmi.stud.vu.lt; 5Baltic Studies Centre, LV-1014 Riga, Latvia; mikelis.grivins@bscresearch.lv

**Keywords:** insect meal, cost and benefit analyses, socio-economic analyses, what-if analyses, circular business models, food leftovers

## Abstract

**Simple Summary:**

The United Nations’ goals for sustainability, like Zero Hunger by 2030, urge us to find new ways to produce food without harming nature. As traditional resources for feed, like soymeal and fishmeal, become scarcer and more expensive, exploring alternative protein sources is one options. Research on using insect meal instead of fishmeal shows promise in terms of animal growth and feed efficiency. This paper assesses the socio-economic benefits and costs of production of insect meal to substitute fishmeal in feed, while also focusing on factors that would affect performance most. The study evaluates the economic value of insect-based products, waste, and greenhouse gas reduction as socio-economic benefits. Data from laboratory trials and case studies on black soldier fly and yellow mealworm reveal negative balances of socio-economic benefits and costs; however, it is possible that upscaling and more effective technologies could move this positively. Changes in nutrient market values could also shift the balance as well as prevent impact on marine ecosystems from reduction fishery; this was not evaluated but could contribute to the overall outcome. Thus, although the current assessment shows a negative balance, it does not necessarily mean that the production of insect meal for feed is not beneficial for society in the long term.

**Abstract:**

Sustainability targets set by the United Nations, such as Zero Hunger by 2030, encourage the search for innovative solutions to enhance food production while preserving the environment. Alternative protein sources for feed, while conventional resources like soymeal and fishmeal become more expensive and scarcer, is one of the possibilities. Studies on substituting fishmeal with insect meal show promising results in terms of animal growth and feed efficiency. This paper aims to assess the socio-economic benefits and costs of insect meal substituting fishmeal in feed and to highlight the factors influencing performance most. The study evaluates the economic value of insect-based products, waste reduction, and reduced greenhouse gas emissions as socio-economic benefits. It combines empirical data derived from laboratory trials and two case studies covering black soldier fly (*Hermetia illucens*) and yellow mealworm (*Tenebrio molitor*). Current analyses reveal negative socio-economic balances, emphasizing that reduction of operating and investment costs through upscaling and technological advancements can give a positive move, as well as factors such as current market valuations for nutrients can change significantly. Thus, a negative balance at the moment does not mean that insect rearing, and larva processing are not desirable from a long-term socio-economic perspective.

## 1. Introduction

Since 2015, when the United Nations (UN) set ambitious sustainability targets, including the achievement of Zero Hunger by 2030, there is still a critical need to work on innovative solutions that would mitigate the strain on traditional food production systems by providing more nutrients without hurting the climate and biodiversity [1]. Within this landscape, the exploration of alternative protein sources remains a strategic way of thinking for businesses operating in the food and agriculture sectors, such as in the Baltic–Nordic region [2]. Existing companies and startups, research institutions, and technology providers are actively seeking partnerships to explore and commercialize alternative protein solutions [3]. By aligning with these emerging trends, enterprises can contribute not only to the realization of UN Sustainability goals but also position themselves as leaders in sustainable and socially responsible practices.

According to nutrient intake and food consumption studies in the Baltic–Nordic region, the intake of pulses, nuts, and seeds has been evaluated as very low in the Nordic and Baltic countries. On the other hand, the mean daily intake of meat and meat products ranged roughly between 100 and 200 g per day, and the mean intake of fish and seafood varied from 80 g per day in Norway to 25 and 35 g per day in Baltic states [4], indicating that the main source of protein in this region is likely to be meat, fish, and seafood products. To maintain this structure of food consumption, the need to find the best solutions for protein sources in feed is pressing for society, consumers, and primary producers in the Baltic–Nordic region, because the costs of conventional feed resources such as soymeal and fishmeal are very high and their availability in the future will be limited due to ongoing climatic changes [5]. Fishmeal produced from forage fish catches and fish processing by-products has a long history of use as an ingredient in both livestock and aquaculture feeding [6,7,8]. For example, in Norway, a total of 1,976,709 tonnes of feed ingredients were used to produce 1,467,655 tonnes of salmon in 2020. The feed was produced from 22.4% marine ingredients, including fish meal [9]. Opinions on the sustainability of wild fish usage in feed differ. On the one hand, it has been stated that only about one-third of the annual world catch of fish is used as raw material for fish meal production, and this is claimed to be a sustainable use for catches for which there is no use in direct human consumption [6]. On the other hand, there are estimates that 90% of wild fish destined for uses other than direct human consumption are food-grade or prime food-grade fish, and this competition with direct human consumption can lead to challenges in food security [10]. Additionally, forage fish catches have been linked to negative impacts on marine ecosystems, including habitat degradation and loss of biodiversity [11,12,13,14].

Numerous studies investigating the possibilities of substituting insect meal for fishmeal in livestock and aquaculture feeds have been carried out in the last decade, showing promising results regarding animal growth performance, nutritional value of the meal, and feed efficiency [13,14]. One significant benefit of seafood as a feed ingredient is its dual provision of protein alongside a rich source of bioavailable micronutrients and essential fatty acids, making it an invaluable contributor to global food and nutrition security [15]. Due to having a complete protein profile and being rich in essential amino acids, insects also represent a valuable nutritional resource [16]. They contain all the essential amino acids required by humans and animals, and their protein content ranges from 35% to 75% dry matter (DM) [17]. Thus, it is reasonable to examine the production of insect-based products not only as a business idea, which could be economically profitable for its investors, but also from a societal benefit perspective, because insects are increasingly recognized as a viable alternative to animal-derived proteins for both animal feed consumption and organic waste management, exemplifying a circular approach to resource utilization [18]. Insects can efficiently utilize organic side streams, such as food production leftovers and food waste, contributing to a more circular and sustainable food system.

The utilization of food production leftovers and food waste in insect meal production offers a higher-value alternative to composting, primarily serving as a substitute for fishmeal (see Figure 1). This approach not only diversifies resource use but also relieves pressure on fisheries facing imminent resource depletion, especially in the future.

Feeding trials indicate that incorporating insect meal into feed yields promising results across various sectors, including salmon farming, as well as pig and poultry husbandry. The study led by Belgit investigated the feasibility of replacing fishmeal with insect meal derived from black soldier fly larvae in the diets of Atlantic salmon. It was found that Atlantic salmon fed with insect meal had comparable growth performance and nutrient utilization to those fed with fishmeal-based diets, indicating that insect meal can be a suitable alternative protein source for fish feed [20]. The key nutritional benefits of using black soldier fly meal in poultry feed include its high protein and fat content, essential for poultry growth and metabolic functions, excellent amino acid profiles, good nutrient digestibility, and positive impact on environmental sustainability [21]. Other research showed that incorporating partially defatted black soldier fly larva meal into diets for weaned piglets does not adversely impact their growth performance, nutrient digestibility, blood profile, gut morphology, or histological features [22]. 

Oil as well as frass is produced as a by-product in the rearing of insects and processing the larvae into insect meal. Frass as a fertilizer could be a sustainable way to recycle nutrients and reduce waste from commercial insect rearing. Frass is a potential organic soil amendment or fertilizer to promote plant growth, as it contains many essential plant nutrients. Moreover, the diet of black soldier flies does have an impact on greenhouse gas emissions from frass applied as fertilizer. Specifically, frass derived from a carbohydrate-based diet caused strong initial N_2_O emissions associated with high N and C availability, while frass derived from a protein-based diet did not have this effect [23].

In general, current studies on insect-based products for feed purposes focus on three main narratives: (1) investigating insects as a solution for organic waste treatment, which not only reduces waste streams but also allows us to valorise nutrients otherwise lost in the value chain; (2) analysing the biochemical composition and nutritional values of insect-based products; and (3) conducting experiments with different insect species to explore the possibilities and outcomes of substituting existing protein sources and evaluating feed efficiency.

Recognizing the valuable product that can be produced in a societally welcomed circular manner, startups are actively pursuing business models to bring insect meal to the market. While business initiatives are still in the early stages, dealing with both technical development and market entry, we took two startup cases in the spotlight to assess the anticipated long-term socio-economic benefits and costs of insect meal production, particularly for substituting fishmeal in livestock and aquaculture feed and to highlight factors having the most significant influence on the results of cost and benefits analyses.

Expected societal benefits, in quantitative terms, so far have been assessed through life cycle analyses and evaluation of greenhouse gas emissions. In such a landscape, our research introduces a novel dimension to the discourse on insect meal utilization. We conducted a comprehensive cost-benefit analysis employing a socio-economic framework, focusing on both the production of insect meal and its potential as a replacement for fishmeal in animal and aquaculture feed. This approach provides valuable insights into the possible socio-economic implications of integrating insect-based products into existing agri-food systems, as well as highlighting the factors that have the most significant impact on the results.

Additionally, we introduced an innovative approach to model insect rearing. Current approaches for modelling insect rearing and its outcomes are mainly based on the use of feed conversion rate (ratio), bioconversion rate, waste reduction, etc. Such approaches are feedstock-specific, which complicates their application and set boundaries because the structure of feed for insects is almost unique in each case. Moreover, these indicators (e.g., feed conversion rate) depend on the fluctuations in the biochemical parameters of insect larvae. To deal with these shortcomings, we developed an approach that involves the following key elements: nitrogen conversion efficiency, nitrogen (N) balance, phosphorous (P) balance, and potassium (K) balance. Although the mass balance approach is not new, it has been mainly used to analyse processes and not to model outcomes.

Our evaluation described in this paper is based on two species of insects—black soldier fly (*Hermetia illucens*, BSF) and yellow mealworm (*Tenebrio molitor*, YM) assuming larva meal as the primary product of insect rearing and larva frass and larva lipids as by-product of this process. We set two main research questions: (1) identifying and assessing in monetary terms the socio-economic benefits and costs associated with insect rearing, larva processing, and larva meal utilization in livestock and aquaculture feed as a substitute for fishmeal and (2) understanding the factors influencing the balance of socio-economic benefits and costs and investigating the what-if situations that could lead to noticeable changes in the overall assessment. 

## 2. Materials and Methods

### 2.1. General Methodology

The study is based on laboratory trials conducted by the Estonian University of Life Sciences and case studies carried out within the research project “Promoting collaboration for sustainable and circular use of bioresources across agriculture, forestry, and aquaculture” (CIRCLE, Project No. EEA-RESEARCH-24) as well as the authors’ modelling and additional research. The laboratory trials were conducted in Tartu (Estonia). These studies yielded fairly detailed data, although this was limited due to the small scale of operations. The data obtained in these trials underlie many technical assumptions and estimates used in the study. The case studies provide the second layer of data. Within the CIRCLE project, we examined two cases:A Lithuanian company (CS1) that rears BSF and processes its larvae into several insect-based products for animal feed and pharmacy;A Norwegian company (CS2) that rears YM and processes its larvae and produces insect-based products mainly to substitute imported sources of proteins and lipids in salmon fish and poultry feed.

Due to confidentiality concerns of the companies, case studies yielded limited information about the cases. Mainly, we derived information about the scale of these cases as well the general aspects of their business models and collaboration environment to create assumptions regarding benefits and costs arising from the implementation of the evaluated circular business initiative. These data from the case studies were combined with the data derived from the laboratory trials to make mathematical models of the cases, implement upscaling, etc. (see below). The mathematical models are based on a simplified model which is composed of two parts—insect rearing and larva processing. The first part involves processing feedstock into larva biomass and frass. The second part involves the processing of larva meal and oil.

The assessment of socio-economic benefits and costs is carried out as a socio-economic analysis (type of cost-benefit analysis) to appraise the cases’ contribution to the welfare of society. The EC guidelines for the cost-benefit analysis [24] is used as the main methodological source. According to these guidelines, the social discount rate of 5% (annual rate) is used for the calculations of the present (discounted) values of benefits and costs. Constant (real) prices (mostly based on prices of 2023/2024) are used to calculate the benefits and costs. Time-averaged prices are used instead of spot prices to mitigate the effects of short-term fluctuations on the assumptions regarding constant prices (see below). In addition to the present value (PV) of benefits and costs, the analysis is supplemented with an equivalent annual annuity (EAA). EAA is calculated by using the standard formula of an annuity. Socio-economic analysis (and cost-benefit analyses in general) is based on an incremental approach, therefore a circular business model (BM) scenario which represents each case study is compared with a counterfactual baseline scenario which is theoretical, assumed by the authors and reflects a typical business practice that would pertain in the absence of circular BM. The choice of the appropriate baseline scenario is a significant element of socio-economic CBA that may affect the results of it. As the implementation of CBMs quite frequently involves a transition away from linear business practices, linear or weakly circular business models were used to define baseline scenarios for each case study. For a more detailed examination of circular BM and baseline scenarios see Section 2.3 and Section 2.4, respectively. 

Socio-economic analysis involves such principles as fiscal corrections, conversion from market to shadow prices, evaluation of non-market impacts, and correction for externalities. Fiscal corrections mainly entail the use of prices without value-added tax (VAT). According to the EC Guidelines, social security payments are considered delayed salary [24]. Therefore, they (including those paid by employers) have been included in labour costs. If not stated otherwise, we assumed conversion factor (SFC) 1.0 to convert financial items into shadow prices (social opportunity costs). The EC Guidelines also suggest that, if an economy is characterised by extensive unemployment or underemployment, SFC may be less than 1 [24]. However, the unemployment rates both in Lithuania and Norway (the 12-month average rate of 7.1% and 3.7%, respectively [25]) cannot be regarded as extensive. Moreover, the EC Guidelines suggest assuming the shadow wage for skilled workers equal or close to the market wage [24]. Thus, we assume that the difference between market wage rates and shadow wage rates is negligible.

The following socio-economic benefits are identified and assessed within the study:Economic value of larva mealEconomic value of larva oilEconomic value of frassAvoidance of organic wasteAvoided GHG emissions from fish meal productionAvoided adverse impact on biodiversity and ecosystems from reduction fishery.

The economic value of larva meal, larva oil, and frass are assessed based on the modelling of larva rearing and processing (see below) and the economic value (social opportunity cost) of protein, lipids (fat), and nutrients (as fertilisers) like N, P, and K. As larva meal is regarded as similar to fishmeal concerning amino acids [26], fishmeal is used as a proxy for the economic value of protein. Although fishmeal contains lipids and some other nutrients as well, we assume that it is priced on a protein basis. The 6-month average price of Peruvian fishmeal (1690 EUR per tonne (t), 65% protein) reported by Indexmundi.com [27] is used to estimate the economic value of protein. The 6-month period allows for mitigating the effect of short-term fluctuations while adequately reflecting the level of price at a base year. In addition, a 20% discount is applied to the fishmeal produced in Lithuania (Baltic countries). Thus, the economic value of larva protein is estimated at 2.60 EUR per kg (Norway) and 2.08 EUR per kg (Lithuania).

It should be noted that in this study, larva oil refers to a product that is obtained during the processing (defatting) of larvae. This product has a very high crude fat content (100% in DM and almost 100% in product). Larva lipids or fat refers to lipids (fat) as nutrients. Biochemical parameters of larva oil (BSF origin) obtained within CS1 indicate that its fatty acid profile differs from that of fish oil. Therefore, we use rapeseed oil (typical plant-based fat) as a proxy for the economic value of lipids. Analogous to the economic value of protein, the economic value is derived from the 6-month average price of rapeseed oil (943 EUR per t) [27] and it is estimated 0.95 EUR per kg of lipids. 

The economic value of frass is calculated as the sum of the economic value of basic nutrients—N, P, and K:(1)$$B_{\mathrm{fr}}=q_{N}\cdot v_{N}+q_{P}\cdot v_{P}+q_{K}\cdot v_{K},$$
where B_fr_ is the economic value of frass (EUR), q_N_ is the output of N in frass (kg), v_N_ is the economic value of N (EUR∙kg^−1^), q_P_ is the output of P in frass (kg), v_P_ is the economic value of P (EUR∙kg^−1^), q_K_ is the output of K in frass (kg), v_K_ is the economic value of K (EUR∙kg^−1^).

The output (amounts) of N, P, and K are derived from mathematical models (see Section 2.2). To estimate the economic values of nutrients, mineral (synthetic) fertilisers are used as proxies, and the statistics of external trade as sources of prices. Ammonium nitrate (Combined Nomenclature (CN) code 31023090) is used to estimate the economic value of N. Monoammonium phosphate (CN code 31054000), which contains both N and P, is used to estimate the economic value of P (v_N_ estimated from ammonium nitrate) by assuming N content 12% and P_2_O_5_ content 52% and applying factor 2.291 (P_2_O_5_-P mass ratio). Potassium chloride (CN code 31042050) is used to estimate the economic value of K. These estimates are based on 12-month average prices of the fertilisers derived from Lithuanian and Norwegian statistics of external trade (see Section 2.3 and Section 2.4). A 12-month period is employed because the prices of fertilisers are influenced by seasonal factors and thus shorter periods can lead to biased estimates.

We assume that feeding insects with food leftovers avoids their becoming waste. Therefore, benefits from avoiding organic waste are assessed by applying the cost of conventional treatment (utilisation) of food leftovers. It is assumed that the tariff (price) on disposal of organic waste reflects the social opportunity cost of the treatment of food leftovers. More detailed estimates are presented in Section 2.3 and Section 2.4.

Fishmeal production (fish processing into fishmeal and fish oil also called reduction from round fish to meal and oil) generates greenhouse gas (GHG) emissions both in the stage of fishing and processing. The avoided GHG emissions are calculated according to the estimated amount of substituted fishmeal and fish oil. We assume that 1 kg of larva protein substitutes the same amount of fishmeal protein. Thus, the substituted amount of fishmeal is estimated by dividing the amount of modelled amount of larva protein by the crude protein content of fishmeal. The crude protein content of 65% (in the product) reported by Indexmundi.com [28] applied. The co-product of fishmeal production is fish oil. Its amount is estimated in two steps. First, the amount of fish is reversely calculated from the estimated substitution of fishmeal by applying the yield of fishmeal reported by SINTEF Ocean [29]. Second, the amount of substituted fish oil is calculated by applying the yield of fish oil reported by SINTEF [29]. Life cycle GHG emission per 1 kg fishmeal and fish oil reported by SINTEF [29] is used to calculate the avoided GHG emissions. The unit cost of CO_2_ 39 EUR per t suggested by the EC guidelines (the central estimate for 2024) [24] is used to assess benefits as well as costs related to GHG emissions.

Benefits from avoiding adverse impact on biodiversity and ecosystems from reduction of fishing arise from a reduced amount of fishing. Thus, our quantitative assessment of avoided reduction of fishing (the amount of fish that corresponds to the substituted fishmeal) outlines the potential of this benefit. However, due to the absence of appropriate and credible unit costs and proxies, we have not assessed these benefits in monetary terms in this study.

The following socio-economic costs are identified and assessed within the study:Investment cost of insect rearingInvestment cost of larva processingEnergy (electricity, heat)Water and sewageConsumablesLabour costGHG emissions from consumed energyNet economic value of substituted fishmeal and fish oil.

The investment cost of insect rearing involves investment in premises (buildings) and equipment. Investment cost estimates are based on the results and estimates of the laboratory trials. In general, the higher the scale, the higher the investment cost. Scale differences can be measured in different ways, e.g., according to feedstock input, larva biomass output, frass output, etc. We use feedstock input in terms of DM because it outlines the investment needs more properly than output indicators. The following non-linear regressions derived from the Technical Report by ADAS and Michelmores [26] is used to estimate the indexes of specific investment needs (premises) or costs (equipment):(2)$$i_{\mathrm{inv}\_\mathrm{pr}}=0.9379\cdot {i_{q\_\mathrm{sc}}}^{-0.26},$$
(3)$$i_{\mathrm{inv}\_\mathrm{eq}}=0.9415\cdot {i_{q\_\mathrm{sc}}}^{-0.269},$$
where i_inv_pr_ is the index of specific investment needs for premises at scale difference or index (i_q_sc_), i_inv_eq_ is the index of specific investment cost for equipment at i_q_sc_, i_q_sc_ calculated as the ratio of feedstock input (in terms of DM) at laboratory scale to feedstock input (at terms DM) at the case level (or modelled level). At the laboratory scale, i_q_sc_ = 1, i_inv_pr_ = 1, and i_inv_eq_ = 1.

Formulas (2) and (3) indicate the exitance of the economy of scale: the increase in scale results in the decrease of specific investment needs or costs. The investment needs for premises (area (m^2^) and volume (m^3^)) are assessed by multiplying the estimated specific investment needs (m^2^ per kg DM feedstock, m^3^ per kg DM feedstock) at the laboratory level, i_q_sc_ and the feedstock input at the case level (or modelled level). To assess the investment cost of equipment, the estimated specific investment cost (EUR per kg DM feedstock) is used. We assume that the investment cost of premises for larva processing is small relative to the investment cost of premises for insect rearing. Therefore, it assumed that the estimated investment of premises for insect rearing also includes the needs for the processing. Formula (3) is also used to assess the investment cost of equipment for larva processing. In this case, i_q_sc_ is calculated as the ratio of larva biomass (in terms of DM). 

We assume that the economic lifetime of premises and equipment is 25 and 5 years, respectively. These are cautious assumptions. As the assumed lifetime of equipment is shorter, four cycles of reinvestment are assumed (at Years 5, 10, 15, and 20, respectively). No residual value is assumed at the end of the economic lifetime. It should be noted that 25 years is also assumed as the reference period of the whole socio-economic analysis.

Energy costs are composed of the cost of electricity and heat. The consumption of electricity in insect rearing is estimated according to feedstock input (in terms of DM) but in larva processing—according to larva biomass (in terms of DM). See Section 2.3 and Section 2.4. for the estimated specific consumptions. The Nord Pool 12-month average price of electricity for Lithuania (area “LT”) is 90.80 EUR per MWh and for Norway (area “NO5”, Bergen region) 57.04 EUR per MWh [30]. The 12-month period is employed due to seasonal factors that affect the prices of electricity. Nord pool data are used to calculate the variable cost of electricity. In addition, the trader’s premium and fixed cost of electricity is estimated. It is assumed that only electricity is used in larva processing. The specific consumption of 2.85 kWh per kg DM larvae is derived from Cámara-Ruiz M. et al. (2023) [31]. Heat needs are assessed by comparing estimated gross heating needs and internal heat gains on a monthly basis. The gross heating need is estimated by the following formula:(4)$$Q_{\mathrm{gr}}=U_{V}\cdot V\cdot \Delta T,$$
where Q_gr_ is the gross heating needs, U_V_ is the specific heat transfer coefficient (W∙m^−3^∙K), V is the volume of premises (m^3^), and ΔT is the indoor/outdoor temperature difference.

Uv is estimated by multiplying insulation value or U-value (W∙m^−2^∙K) and shell surface/volume ratio (SV ratio). According to the report by Kemna R., we estimated the U-value 0.392 W∙m^−2^∙K (estimated based on the U-values of efficient buildings erected after 2006) and assumed SV ratio 0.33 (the EU average of industrial buildings) [32]. Thus, Uv is estimated as 0.129 W∙m^−3^∙K. ΔT is estimated by assuming an indoor temperature of 28 °C and using the monthly average outdoor temperatures reported by Climate-data.org for Tartu (the laboratory trials), Vilnius (CS1), and Bergen (CS2) [33].

Internal heat gains are composed of metabolic heat and heat gain from ventilation. Metabolic heat is related to the difference between the gross energy of feedstock and the gross energy of larva biomass and frass. The gross energy of feedstock is calculated according to its biochemical composition and by applying values of 5.65 kcal∙g^−1^ (23.6 MJ∙g^−1^), 9.45 kcal∙g^−1^ (39.5 MJ∙g^−1^), and 4.2 kcal∙g^−l^ (17.6 MJ∙g^−1^) for protein, lipids, and carbohydrates, respectively [34]. The heat gain from ventilation is related to the fact that electric energy consumed by supply ventilation largely transforms into kinetic energy of airflow which, in turn, transforms into heat due to friction, etc. In the laboratory trials, the ventilation consumed about 85% of all the consumption of electricity. However, it should be noted that this ventilation was powerful concerning the volume of premises, it had a preheating unit, and there was only supply ventilation. Regarding CS1 and CS2, we assume that the ventilation will account for 60% of the consumption of electricity. It is also assumed that there are both supply and exhaust ventilation and they account for 50% and 50%, respectively. In addition, it is assumed that about 70% of the consumption of the supply ventilation makes the internal heat gain. Thus, it is estimated that the heat gain from ventilation accounts for about 21% of the total consumption of electricity.

Water and sewage costs are composed of water consumption, which does not generate wastewater (mainly water for the preparation of feed), and water consumption, which generates wastewater (water for washing, etc.). The water consumption is estimated according to the amount of feedstock (in terms of DM). The tariffs (prices) of water supply companies are applied to calculate these costs. For more information, see Section 2.3 and Section 2.4. Consumables include operating (material and service) costs that are not energy, water/sewage, and labour. These costs are assumed proportional to the cost of electricity (both variable and fixed). The estimates at the laboratory scale (the ratio to the cost of electricity) are used as the basis for the assessment.

Labour cost is assessed by estimating the required person-hours and assuming the unit cost of labour. The following non-linear regressions derived from the Technical Report by ADAS and Michelmores [26] is used to estimate the index of specific labour consumption:(5)$$i_{L}=1.1142\cdot {i_{q\_\mathrm{sc}}}^{-0.279},$$
where i_L_ is the index of labour specific consumption at i_q_sc_. At the laboratory scale, i_L_ = 1.

Formula (5) indicates the economy of scale as i_L_ decreases if the scale increases. The required person-hours are estimated by multiplying the specific labour consumption (person-hours per 1 t DM feedstock) at the laboratory scale, i_L_, and the input of feedstock at the case level (or modelled level). According to Eurostat (average labour cost levels), we assume the unit cost labour (including employer’s taxes) as 14.7 EUR per hour in Lithuania and 51.9 EUR per hour in Norway [35].

GHG emissions from consumed energy is calculated according to the consumption of electricity in insect rearing and larva processing. GHG emissions factors (activity-based approach) for Lithuanian electricity (0.079 t CO_2_-eq per MWh) and for Norwegian electricity (0.012 t CO_2_-eq per MWh), as reported by the Joint Research Centre [36] are used in calculations. The same CO_2_ unit cost suggested by the EC guidelines is applied (see above).

The net economic value of substituted fishmeal and fish oil refers to the net socio-economic value of fishmeal and fish oil which the society loses if fishmeal is substituted. The substituted amounts, which make up the avoided GHG emissions from fishmeal production, also contribute to these socio-economic costs. Earnings before interest and taxes (EBIT) is used as a proxy for this net economic value. We regard EBIT as the more appropriate proxy than value added because labour cost is included in the socio-economic costs of the circular BM scenario and labour is resource for which social opportunity cost is not zero. The net value is assessed by multiplying the price of fishmeal and fish oil and the ratio of EBIT to turnover (net sales). Due to the specifics of Lithuania, it was not possible to obtain the data regarding the EBIT/turnover ratio. Therefore, the financial statements of Latvian producers were explored. According to the financial data from one Latvian company that processes fish into meal and oil and has almost no other business, the average EBIT/turnover ratio was 15.6% in the period 2017–2022. It should be noted that some positive net value is probably generated in fishing stage as well. However, the ratio of 15.6% includes investment subsidies which the company received and recognised in revue. According to the general principles of socio-economic analysis, these subsidies are transfers that should be excluded. The average ratio of these subsidies to turnover was 2.3% in the period 2017–2022. We assume these 2.3% approximately reflect the net economic value generated in the fishing stage. Therefore, it is assumed that the ratio 15.6% also includes the net economic value for the whole chain (both fishing and processing) in Lithuania. It was also challenging to obtain data regarding EBIT/turnover ratio in Norway as many companies are consolidated and involved in various businesses. The analysis of one Norwegian fishmeal producer revealed that the average ratio in period 2019–2022 was 9.7%. By assuming the ratio of raw material cost (purchasing of fish) to turnover 40% (very approximate assumption) and the smaller EBIT/turnover ratio in fishing (6.0%), to aggregated EBIT/turnover ratio is estimated 12.1% in Norway.

What-if analysis was conducted to reveal the impact of changes. Several situations, which included so-called “no change” situation (no changes in the scale, specific costs, or other variables of the case) and changes in some key variables, were examined. For more detailed information about examined situations, see Section 2.3 and Section 2.4.

### 2.2. Modelling Insect Rearing and Larva Processing

A typical approach for modelling insect rearing involves operating with a feed conversion rate (FCR) [26,37]. Some researchers operate with partially similar indicators, e.g., bioconversion rate [38], waste reduction [39], the waste conversion efficiency [40], etc. However, FCR and other mentioned indicators have two significant drawbacks, even if calculated on a DM basis. First, these indicators are heavily affected by the type and composition of feedstock (substrate). They are feedstock specific, which limits their application. Second, these indicators are also affected by the fluctuations in biochemical parameters (e.g., crude protein content, crude fat content) of larvae. Gold et al. also offers to use protein conversion efficiency (PCE) [38]. This indicator addresses these shortcomings. However, it has a specific shortcoming; namely, it depends on nitrogen-to-protein conversion factors applied to both feedstock and larvae. It is a significant weakness of PCE, particularly if conversion factors applied to feedstock and/or larvae are not disclosed (known). Although many scholars use the standard (generic) factor 6.25 (e.g., INRA [41]), other factors are also used, e.g., 5.7 for some cereals [42], 4.67 for BSF larvae [38].

To address these challenges, we have introduced a new indicator to model insect larva rearing—nitrogen conversion efficiency (NCE):(6)$$\mathrm{NCE}=\frac{N\_{\mathrm{output}}_{\mathrm{larvae}}}{N\_{\mathrm{input}}_{\mathrm{feed}}},$$
where N_output_larvae_ is the N output by larva biomass or larvae N (kg N or other mass unit), N_input_feed_ is the N input by feedstock or feedstock N (kg n or other mass unit).

We model the biomass of larvae by multiplying the N input by feedstock and NCE. The estimated N content of feedstock is presented in Section 2.3 and Section 2.4. The analysis of the data from the research by Gold et al. (2020) [38] indicates that NCE is very stable regarding plant-based substrates as feedstock for BSF—47.8% (poultry feed as feedstock), 48.9% (vegetable canteen waste), and 50.1% (mill by-products). By applying the assumed N content of larvae (see Section 2.3 and Section 2.4), larva biomass (in terms of DM) is calculated from N_output_larvae_. This larva biomass is used to calculate P and K outputs (see below).

We use the N balance approach and model the whole outcomes of insect rearing according to the following equitation:(7)$$N\_{\mathrm{input}}_{\mathrm{feed}}=N\_{\mathrm{output}}_{\mathrm{larvae}}+N\_{\mathrm{output}}_{\mathrm{frass}}+N_{\mathrm{loss}},$$
where N_output_frass_ is the N output by frass or frass N (kg N), and N_loss_ is the gaseous losses of N (kg N), e.g., as ammonia (NH_3_), nitrogen oxides (NO_x_), etc. 

N_loss_ is calculated using the following formula:(8)$$N_{\mathrm{loss}}=k_{\mathrm{loss}}\cdot N\_{\mathrm{input}}_{\mathrm{feed}},$$
where k_loss_ is the share of the N input that is lost. 

Thus, N_output_frass_ is calculated by subtracting N_output_frass_ and N_loss_ from N_input_eed_. We also use P and K balance approaches to model the outcomes:(9)$$P\_{\mathrm{input}}_{\mathrm{feed}}=P\_{\mathrm{output}}_{\mathrm{larvae}}+P\_{\mathrm{output}}_{\mathrm{frass}},$$
(10)$$K\_{\mathrm{input}}_{\mathrm{feed}}=K\_{\mathrm{output}}_{\mathrm{larvae}}+K\_{\mathrm{output}}_{\mathrm{frass}},$$
where P_input_feed_ and K_input_feed_ are the P and K input by feedstock, respectively, P_output_larvae_ and K_output_larvae_ are the P and K output by larva biomass, respectively, and P_output_frass_ and K_output_frass_ are the P and K output by frass, respectively. No losses of P or K are assumed.

P_input_feed_ and K_input_feed_ are calculated according to the estimated P and K contend of feedstock. P_output_larvae_ and are K_output_larvae_ are calculated according to the calculated larva biomass (in terms of DM) and the assumed P and K contend of larvae. The assumptions about and estimates of P and K contents are presented in Section 2.3 and Section 2.4. P_output_frass_ (also K_output_frass_) is calculated as the difference between P_input_feed_ (K_input_feed_) and P_output_larvae_ (K_output_larvae_).

Larva processing is modelled by assuming no considerable loss of DM, protein (and N), or lipids during the process. Therefore, it is assumed that all the protein (and N) of larva biomass flows to larva meal. It is also assumed that all the DM of larva oil is composed of lipids (crude fat content 100% in DM). Thus, the lipids and DM of larvae are allocated between larva meal and oil. The crude fat content of larva meal varies, depending on the technology of processing (defatting) and other factors. According to Eide et al. (2024) [43], the crude fat content is assumed 5.7% (in DM). The mass of lipids allocated to larva oil is calculated by the following formula:(11)$$M_{f\_\mathrm{larvae}\_\mathrm{oil}}=\frac{{\mathrm{DM}}_{\mathrm{larvae}}\cdot \left({\mathrm{CF}}_{\mathrm{larvae}}-{\mathrm{CF}}_{\mathrm{larvae}\_\mathrm{meal}}\right)}{\left(1-{\mathrm{CF}}_{\mathrm{larvae}\_\mathrm{meal}}\right)},$$
where M_f_larvae_oil_ is the mass of lipids allocated to larva oil (kg), DM_larvae_ is the DM of larva biomass (kg), CF_larvae_ is the crude fat content of larvae (decimal fraction), CF_larvae_meal_ is the the crude fat content of larva meal (decimal fraction). M_f_larvae_oil_ equals to the DM of larva oil. Thus, the mass of lipids allocated to larva meal is calculated by subtracting M_f_larvae_oil_ from the mass of lipids in larvae (DM_larvae_∙CF_larvae_). The DM of larva meal is calculated as the difference between DM_larvae_ and M_f_larvae_oil_.

### 2.3. Case study of Lithuania: Sustainability with Insects

The case study of Lithuania (CS1) represents a small start-up company established by organic waste experts with a main goal of returning food waste back to the food chain using black soldier fly [44]. The business idea is to create value by transforming food leftovers into valuable proteins, lipids, and fertilizers. Thus, this circular business initiative provides a solution that addresses at least two environmental concerns: reduction of waste and the growing interest in alternative protein sources [44]. Rearing and processing of insects in this case serves as a pivotal link to improve circularity in the bio-resource flow among agriculture, food processing, and organic leftovers management if compared to theoretical baseline scenario (see Figure 2). 

The bio-resource flow described in the scenario of circular BM (Figure 2) follows a closed-loop, circular practice integrating crop production, animal husbandry, food processing, and BSF rearing. Initially, feed crops cultivated in crop production serve as primary inputs for feed in animal husbandry, facilitating the production of meat and other animal products. The resulting manure is then recycled back into crop production, where it acts as a fertilizer, replenishing soil fertility and supporting continued crop growth. This cyclic relationship forms a traditional loop between crop production and animal husbandry, optimizing nutrient cycling and minimizing the necessity for purchased synthetic fertilisers. Subsequently, products derived from the primary sectors, including various crops and meat are directed to the food processing sector. Here, organic leftovers generated during food processing are diverted to the facility of black soldier fly rearing and larva processing. Within the black soldier fly rearing process, organic waste is utilized as a substrate for black soldier fly larvae production. The larvae undergo processing to extract nutrients such as proteins and lipids, which are then incorporated into animal feed formulations, substituting for fishmeal. Additionally, the residual organic matter, called larvae frass, is utilized as a fertilizer supporting crop production.

In the baseline scenario, we assumed that the same system without the insect rearing stage involved would operate more linearly—the interconnection between crop production, animal husbandry, and food processing would remain, however, the linear bio-resource flow would operate on a “take–make–dispose” model and nutrients that still are present in organic leftovers would not be recognized, valorised and returned to the production chain. Blue arrows in Figure 2 indicate the differences between the flow of bio-resources in circular and baseline BM scenarios; red arrow indicates the threats to nature. According to those, the key points for evaluation of socio-economic benefits and costs associated with the implementation of circular BM in CS1 are related to the introduction of new insect-based products, e.g., larva meal, lipids, and frass in the system mainly to substitute fish meal component in animal husbandry and to avoid organic waste.

According to the company’s provided information, the company processes 200–500 kg of food leftovers (wet weight) per day. Thus, the estimated annual amount is 87,500 kg (wet weight). Based on the assumed DM content 51.8% [45], the input of feedstock is calculated at 44,800 kg DM per year. The biochemical parameters of feedstock are estimated by assuming a mix of carrot, onion, and potato (equal share in the mix) and using these products as proxies. Considering the biochemical parameters of these products, reported by Eatthismuch.com [46], the main parameters of feedstock are estimated as follows: N content 1.76% (in DM), P content 0.29% (in DM), and K content 2.07% (in DM). According to INRA, the following biochemical parameters of larvae are assumed: crude protein 41.1% (in DM), N content 6.58% (in DM), crude fat content 35.5% (in DM), P content 0.72% (in DM) and K content 0.76% (in DM) [41]. To estimate the need for investment and labour, the data derived from the laboratory trials (Tartu) were used and upscaled to the scope of CS1 (the estimated increase was 9.9 times). Key estimates and assessments are presented in Table 1.

According to Tambone et al. (2010), N losses during anaerobic digestion (currently a typical method of organic waste utilisation) of food residues typically vary within the range of 10–50% [47]. Considering that estimated N_loss_ falls almost in the middle of this range, it is cautiously assumed that N losses in the circular BM scenario and the baseline scenario are equal. Therefore, it is assumed that N pollution is neither generated nor avoided. The system of separate food and kitchen waste collection was introduced in Lithuania only in 2024. This system is under development, which renders it challenging to assess the social opportunity cost of organic waste treatment. The tariff for organic waste management in the Alytus region 60.50 EUR per t (wet weight) [48] or 118.16 EUR per t DM (estimated dry matter cost) is assumed as the social opportunity cost of the treatment of food leftovers. Based on the external trade (import) statistics of Lithuania [49], the v_N_, v_P_ and v_K_ are estimated at 812 EUR per t, 1998 EUR per t, and 686 EUR per t, respectively.

The specific investment cost of premises is assumed 1200 EUR per m^2^. Based on from Cámara-Ruiz M. at al. (2023) [31], the specific consumption of electricity is estimated 0.333 kWh per kg DM feedstock, water consumption (without sewage)—1.19 m^3^ per t DM feedstock, water consumption (with sewage)—0.73 m^3^ per t DM feedstock. Fixed cost of electricity estimated 0.7% from investment in equipment. Based on “Vilniaus vandenys” tariffs (prices), the cost of water is assumed 0.86 EUR per m^3^ (without sewage) and 1.74 EUR per m^3^ (with sewage) [50]. Consumables are estimated 152.7% of the cost of electricity (both variable and fixed). According to Parodi et al. (2020), is assumed that metabolic heat accounts for 23% of the gross energy of feedstock [51]. It should be noted that calculated net heat need is negligible at the scale of the laboratory (a small need only in some winter months in Tartu) but internal heat gains substantially exceed gross heating needs at the scale of CS1.

The following situations were examined in what-if analysis:CS1A—no changes in assumptions or estimatesCS1B—50-fold increase in the scale, other assumptions and estimates the same as of CS1ACS1C—50-fold increase in the scale and increased NCE (48.9%)CS1D—50-fold increase in the scale and consumables reduced by 50%CS1E—50-fold increase in the scale and labour intensity reduced by 50%CS1F—50-fold increase in the scale and investment intensity reduced by 50%CS1G—50-fold increase in the scale and larva oil valued as fish oil (the price of feed-grade fish oil derived from the external trade statistics of Norway [52], 20% discount applied)

### 2.4. Case Study of Norway: Insects for Circular Economy

The case study of the Norwegian company (CS2) represents a start-up specializing in yellow mealworm production from residual raw materials and former food. The company has developed robust production technologies and built a fully operational pilot factory using methods that are ready to be scaled up [44]. The business idea is based on three main elements: valorisation of food waste and residual raw materials; automized production process; and products for sectors and actors who are willing to pay a premium for circular products with a low environmental footprint. Flexibility regarding markets can be maintained because once the larvae have been produced, they can go to food, food ingredients, or feed products however the company aims to address local salmon fish, and poultry farms providing them with Norwegian-produced sources of protein and micronutrients to substitute imported fishmeal and soybean meal [44]. The cultivation and processing of insects in this case greatly enhance the bioresource cycle, drawing in new contributors and maximizing the circulation of nutrients within the system (see Figure 3).

The flow of bio-resources described in the scenario of circular BM (Figure 3) involves a symbiotic relationship between the local waste management company, horticulture farms, and a YM rearing and processing company. Collected residual raw materials from horticulture farms and sorted organic waste from the local waste management company become a feed for YM larvae to extract nutrients such as proteins, fats, and micronutrients out of organic waste which otherwise (baseline scenario) would be composted and returned to local farms as fertiliser, incinerated or landfilled. It is assumed in this case that baseline scenarios primarily focus on waste disposal through composting, landfilling, or incineration, resulting in the loss of potential resources and nutrients contained within the organic leftovers and waste. In the circular BM scenario, no bio-resources are lost—what does not become larvae becomes fertilizer. Blue arrows in Figure 3 indicate the differences between the flow of bio-resources in circular and baseline BM scenarios; red arrow indicates the threats to nature. According to those we have outlined the key points for evaluation of socio-economic benefits and costs associated with the implementation of circular BM in CS2 which are related to the introduction of new insect-based products, e.g., larva meal, lipids, and frass in the system mainly to substitute fish meal component in poultry and salmon fish feeding and to avoid organic waste.

According to the company’s reported information, its target amount of production is 70 t larvae (wet weight) per week. By assuming the DM content 36% (derived from Liu C. et al. (2020) [53], the amount is estimated 25.2 t DM larva biomass per week. This amount is about 2150 higher than the scale of the laboratory trials (see Table 2). The biochemical parameters of feedstock are estimated by assuming a mix of wheat bran (45% share in the mix), oat (45%), carrot (5%), and potato (5%), and using these products as proxies. Considering the biochemical parameters of these products, reported by Eatthismuch.com [46], the main parameters of feedstock are estimated as follows: N content 2.41% (in DM), P content 0.78% (in DM), and K content 0.88% (in DM). According to INRA, the following biochemical parameters of larvae are assumed: crude protein 50.4% (in DM), N content 8.06% (in DM), crude fat content 35.6% (in DM), P content 0.77% (in DM) and K content 0.95% (in DM) [41]. To estimate the needs for investment and labour, the data derived from the laboratory trials (with some adjustments, see Table 2) were used and upscaled to the scope of CS2 (the estimated increase—2150 times). Key estimates and assessments are presented in Table 2.

Although the assumed N_loss_ is 15% higher than in the case of BSF, it still falls in the middle of this range reported by Tambone et al. (2010) [47]. Therefore, no socio-economic benefits or costs regarding N pollution are assessed. Organic waste treatment (utilisation) in Norway is more developed than in Lithuania. In the Bergen region, the cost of utilisation of separated food waste is practically zero [44]. Thus, the social opportunity cost of the treatment of food leftovers is also assumed zero. Based on the external trade statistics of Norway [52], the v_N_, v_P,_ and v_K_ are estimated 1718 EUR per t, 3185 EUR per t and 2332 EUR per t, respectively. The euro foreign exchange reference rates published by the European Central Bank (ECB) [54] are used to convert values from Norwegian Krone (NOK) to EUR. 

Based on Boligfiks.no [55] the specific investment cost of premises is assumed at 2700 EUR per m^2^. The specific consumption of electricity is estimated 0.383 kWh per kg DM feedstock (15% more than BSF), water consumption (without sewage)—1.01 m^3^ per t DM feedstock (15% less than BSF), water consumption (with sewage)—0.62 m^3^ per t DM feedstock (15% more than BSF). The fixed cost of electricity is estimated 0.7% from investment in equipment (the same as BSF). According to Voss Municipality water supply tariff tariffs (prices), the cost of water is assumed 1.98 EUR per m^3^ (without sewage) and 4.11 EUR per m^3^ (with sewage) [56]. Consumables are estimated 129.8% of the cost of electricity (15% less than BSF). Due to uncertainty about the metabolic heat gains in YM rearing, it was assumed that the metabolic heat is 50% less than in BSF rearing, i.e., 11.5% of the gross energy of feedstock. Despite assumed lower internal heat gains from metabolic heat, the calculations indicate negligible net heed need only in winter months (in Tartu). At the scale of CS2 internal heat gains substantially exceed gross heating needs.

The following situations were examined in the what-if analysis:CS2A—no changes in assumptions or estimatesCS2B—consumables reduced by 50%CS1C—labour intensity reduced by 50%CS1D—investment intensity reduced by 50%CS1E—larva oil valued as fish oil (the price of feed-grade fish oil derived from the external trade statistics of Norway [52])CS1F—10-fold increase in the scale

## 3. Results

Based on the methodology, the data, and the assumptions described above, the socio-economic benefits and costs are assessed for both cases (CS1 and CS2). In addition, what-if analysis is conducted to examine all the defined situations (see Section 2). The results of CS1 (including what-if analysis of the main situations) are presented in Table 3. The results of all the analysed situations are presented in Appendix A. Detailed calculations are available in Appendix A.

The results of CS1 indicate the balance of socio-economic benefits and costs are generally negative at current assumptions and estimates. However, the results of what-if analysis suggest that the results tend to improve if the scale is substantially increased (e.g., 50 times). Nevertheless, additional improvement in cost-efficiency is required to achieve a positive balance without considering investment. Situation CS1D, which involves reduced cost of consumables (see Section 2.3), demonstrates the positive balance excluding investment. The other situations with increased scale indicate the negative balance both excluding and including investment (see Appendix A). It should be noted that increased NCE (situation CS1C) does not improve the balance per se because the increase in the benefits is not sufficient to offset the increase in costs caused by a higher yield of larva biomass. The what-if analysis examines only limited situations for CS1. However, it outlines that combined improvements (e.g., reduced cost of consumables, reduced labour intensity, reduced investment intensity) could result in more significant improvement in the balance between socio-economic benefits and costs. 

The results of CS2 (including what-if analysis of the main situations) are presented in Table 4. The results of all analysed situations are presented in Appendix A. Detailed calculations are available in Appendix A.

The results of CS2 are similar to the results of CS1, as they indicate the negative balance of socio-economic benefits and costs at current assumptions and estimates. The results of what-if analysis are also similar. They reveal that a significant reduction of consumables or labour intensity results in the positive balance excluding investment. Unlike CS1 (namely situation CS1G), the increased valuation of larva oil makes the balanced excluding investment positive (situation CS2E). The 10-fold increase in scale (situation CS2F) also results in the positive balance excluding investment. However, such an increase in scale could probably be challenging as the scale of situation CS2A, which involves the input of food leftovers 9746.0 t DM per year, is quite high. Nevertheless, the balance including investment remains negative even at this increased scale.

The assessed amounts of avoided fishing (see Appendix A) imply that avoided adverse impact on biodiversity and ecosystems from reduction fishery very likely contribute to the overall balances of socio-economic benefits and costs substantially. According to our findings, 1 t DM feedstock treated by BSF allows avoiding fishing by 0.36 t (if NCE assumed 39.9%) or even 0.44 t (if NCE assumed 48.9%), but the treatment by YM allows avoiding fishing by 0.52 t per 1 t DM feedstock (if NCE assumed 45.0%). However, it should be noted that the feedstocks have different biochemical parameters (see Section 2.3 and Section 2.4). Further research is necessary to quantify these benefits in monetary terms.

## 4. Discussion

Insect meal is a promising fishmeal substitute and a high-quality source of nutrients [43,57]. Recent EU legislation approving its use in feed production marks a significant step forward [58]. Studies have documented the benefits of incorporating insects into fish diets, demonstrating their potential as a viable alternative to traditional protein sources [59,60,61].

This study highlights the potential socio-economic benefits associated with insect meal as a fishmeal substitute, and it is based on the operational scale and circular business models implemented in practice in Lithuania and Norway; however, their locality is not decisive and can be attributed to similar business initiatives in the Baltic–Nordic region. 

Based on scientific publications, we argue that integrating insects into innovative business models has the potential to enhance the environmental, social, and economic performance of agri-food systems [62] our assessment focused on the monetary evaluation of those potentials. However, quantifying biodiversity and ecosystem benefits in monetary terms remains a challenge that necessitates further investigation. It has been stated that fishmeal production is mainly sourced from forage fish species [63]. Forage fish are important conduits of energy transfer in food webs for many marine ecosystems [10,12]. By promoting the use of insects as a feed ingredient, pressure on wild fish stocks and other natural resources can be reduced, helping to conserve biodiversity and ecosystem health [62]. The reduction of demand for fishmeal and alternative feed sources allows us to safeguard marine ecosystems and the valuable services they provide. Thus, it can be logically assumed, there is a place for additional monetary benefits. It has also been stated that the utilization of insects as fish feed in aquaculture presents a multifaceted solution, offering potential fishmeal substitution, efficient utilization of organic waste, and environmental benefits such as reduced natural resource depletion, biodiversity loss, and pollution levels [17].

The findings indicate the negative balance of socio-economic benefits and costs at current assumptions and estimates and suggest that reducing both operating and investment socio-economic costs is necessary to achieve a positive balance between socio-economic benefits and costs. Our assumptions and estimates could be too cautious, which affects the assessment of socio-economic costs, and we did not find other studies indicating socio-economic benefit and cost assessment results to compare our assessment with. However, there are some other studies indicating that BSF is more expensive than fishmeal [43], and it is important to consider factors such as scalability, production efficiency, and market dynamics that could influence the cost competitiveness of insect-based feeds [9,26]. Additionally, the life cycle analysis led by Thevenot suggests that the cumulative environmental impact of YM meal, including factors like energy consumption and CO_2_ emissions, exceeds that of soybean meal or fish meal when considering 1 kg of protein [64]. While we may not align with every conclusion drawn in this study, it does bear some semblance to our outcomes.

Overall, the findings suggest that reducing both operating and investment socio-economic costs is necessary to achieve a positive balance between socio-economic benefits and costs. Thus, upscaling and developments in technology to reduce labour intensity, investment intensity, etc., are the most likely pathways. It should also be noted that the valuation of insect-based protein and lipids, as well as nutrients (N, P, K) in frass, substantially affects the assessment of socio-economic benefits. Our results are derived from current market valuations of fish-based protein, plant-based lipids, and mineral fertilizers. The mounting challenges regarding the availability of marine resources and mineral (synthetic) fertilizers will very likely result in higher social opportunity costs of protein, lipids, and nutrients (N, P, K) in the future. Therefore, these negative balances do not necessarily mean that insect (BSF and YM) rearing and larva processing are not desirable from a long-term socio-economic perspective.

Future research will have to address the valuation of insect-based feed (and probably food as well) ingredients in more detail as, e.g., larva meal does not contain only protein. One of the options how to address this issue is offered by the concept of economic nutrient units (ENU) proposed by Auzins et al. (2021) [55]. ENU concept allows comparing and valuing feed ingredients with different biochemical parameters as well as it allows considering amino acid profile or even fatty acid profile. However, this concept has been mainly applied to plant-based feed ingredients. Some refinement is likely required to apply this concept to animal-based (including insect-based) feed ingredients.

Political initiatives can be a driver that brings changes into the system and thus affect the evaluation assumptions and results. In the Norwegian context, there are political developments that are supposed to affect the future of alternatives to fishmeal and other non-circular imported feed proteins. In 2023, the Ministry of Trade, Industry, and Fisheries introduced sustainable feed as a national social mission. A main part of this mission is a new research agenda managed by the Research Council of Norway, and future research grant calls reflecting this overall national ambition/mission [65]. This mission aligns perfectly with the challenges identified in our study. The goal of reducing reliance on non-circular feed proteins aligns with the need to minimize reliance on fishmeal, while the focus on greenhouse gas reduction aligns with the broader environmental benefits associated with alternative feed sources.

However, we must acknowledge the ongoing challenges associated with insect meal production. Challenges persist in improving farm performance, given variations in production processes, operational scale, insect species reared, and regulatory constraints faced by farmers [18]. Technical hurdles, economic considerations, and even consumer acceptance all play a role [66,67,68]. Addressing these challenges will require continued research and innovation, alongside efforts to educate consumers about the environmental (especially biodiversity and ecosystem) and nutritional benefits of insect-based feeds.

## 5. Conclusions

The production of insect meal as an alternative protein source in feed has so far been presented in studies as a solution with a potential to enhance the environmental, social, and economic performance of agri-food systems. It has been said that utilizing insects as fish feed is seen as a multifaceted solution, offering opportunities for expanding the diversity of protein sources, efficient utilization of organic waste, and environmental benefits such as reduced natural resource depletion, biodiversity loss, and pollution levels. In our research, we conducted socio-economic benefit and cost analyses for two case studies, considering their circular business model and operational scale. We assumed that insect meal processed within those cases would replace fishmeal in aquaculture and livestock feed. The assessment reveals a negative socio-economic cost and benefit balance at current assumptions and estimates. The study highlights the importance of scaling up and reduction of operational and investment costs, which could ultimately lead to a more positive outcome. Therefore, upscaling and technological advancements aimed at reducing labour and investment intensity are likely the most promising pathways for further development.

It is important to note that the current assessment does not include monetarily assessed benefits arising from the avoided adverse impacts on biodiversity and ecosystems resulting from the reduction of fishery, which are likely of significant importance, due to the current trends in the development of wild fishing and aquaculture. Further research is needed to estimate these benefits accurately.

The valuation of the economic value of insect-based products also emerges as an important factor influencing socio-economic assessment. Our study highlights the significant impact of valuing insect-derived protein, lipids, and nutrients, such as nitrogen, phosphorus, and potassium in frass, for a better understanding of their socio-economic benefits. Our current assessment is based on market valuations of fish-based protein, plant-based lipids, and mineral fertilizers. However, mounting challenges regarding the availability of marine resources and mineral fertilizers will likely result in higher social opportunity costs for similar nutrients in the future. Other researchers have also noted that the main challenges with novel ingredients produced from organic waste or food processing by-products are the cost of the products and scaling up production to significant amounts. The coming years will determine whether these ingredients can be produced in sufficient quantities and at a low enough cost to be used in feed.

Policy initiatives such as Norway’s sustainable feed mission signal a broader shift towards supporting alternative protein sources and can facilitate further research and innovation to address scalability issues, reduce production costs, and enhance consumer confidence in insect-derived feed ingredients. Continued investment in research and development is essential to overcome these hurdles and unlock the theoretical potential of insect-based feed solutions in practice.

Moving forward, collaboration between stakeholders across the value chain, including producers, researchers, policymakers, and industry players, will be instrumental in driving progress toward insect-based feed solutions. Case studies assessed show that insect rearing and larva meal processing improves the circularity of bio-resources and closes the loop in collaboration with other stakeholders involved in agri-food system.

## Figures and Tables

**Figure 1 animals-14-01461-f001:**
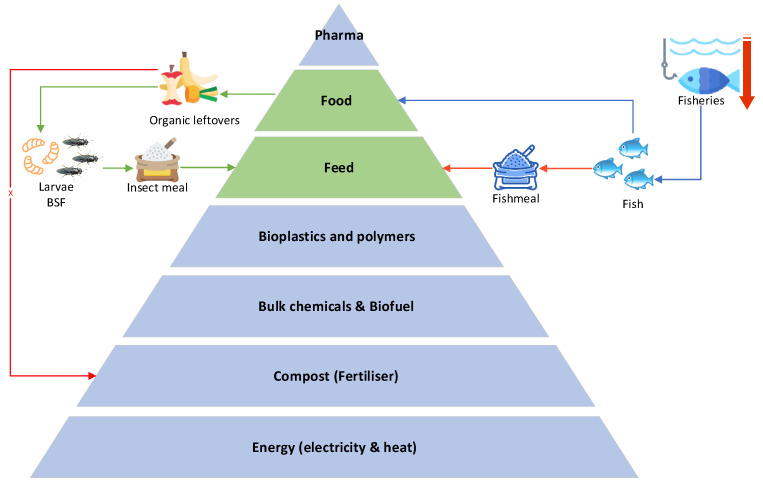
Biomass value pyramid (modified [19]). Focus on fishmeal substitution with insect meal, adding value to organic leftovers.

**Figure 2 animals-14-01461-f002:**
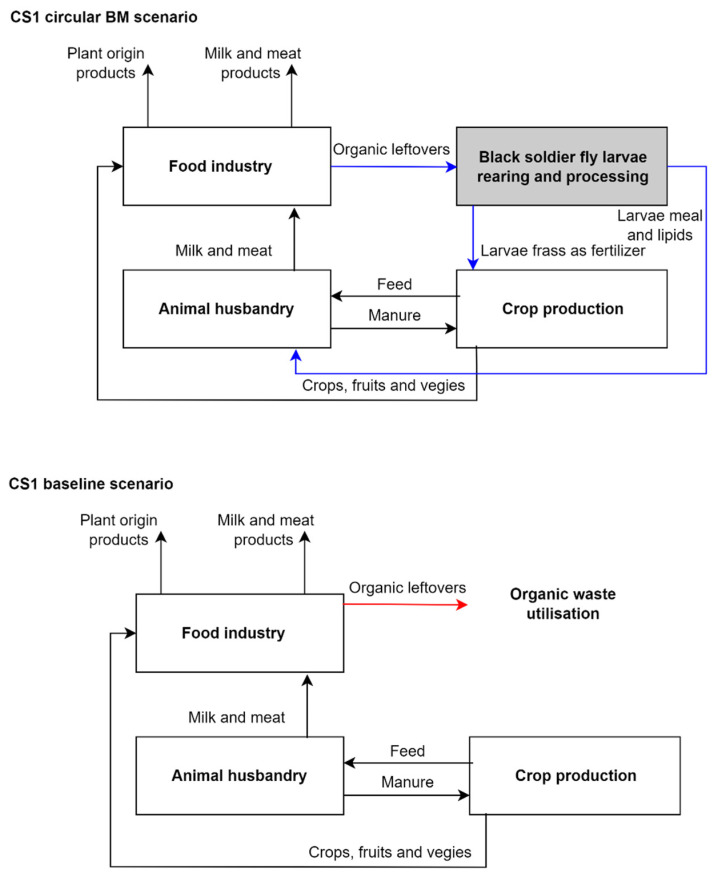
The flow of bio-resources in circular and baseline BM scenarios for CS1. Blue arrows indicate the differences between circular and baseline BM scenarios. The red arrow indicates threats to nature. Source: construction created by authors.

**Figure 3 animals-14-01461-f003:**
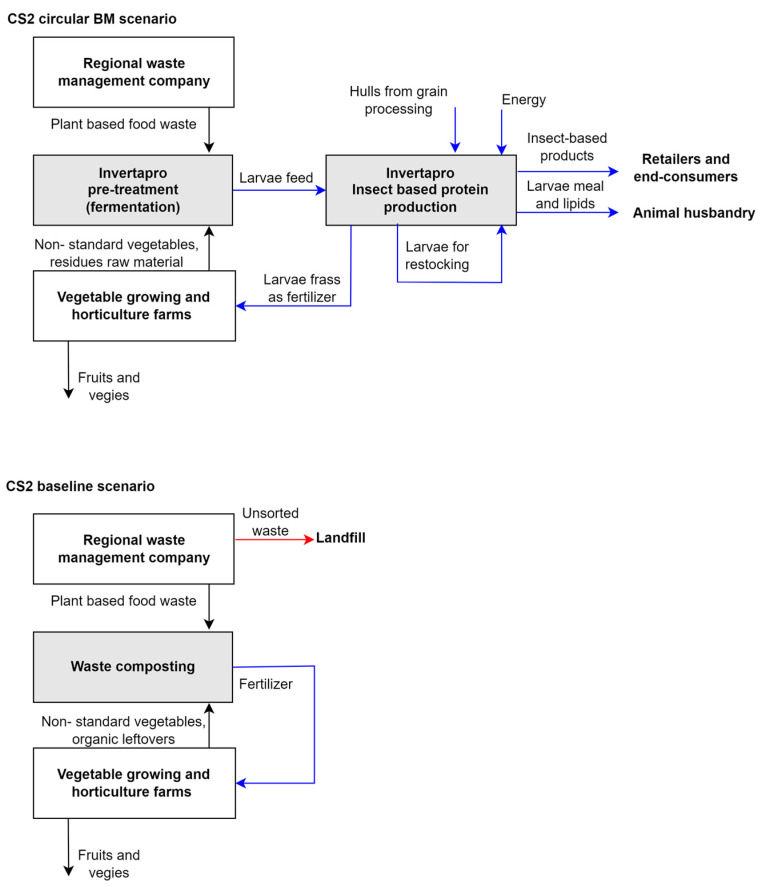
The flow of bio-resources in circular and baseline BM scenarios for CS2. Blue arrows indicate the differences between circular and baseline BM scenarios. Red arrow indicates threats to nature. Source: construction created by authors.

**Table 1 animals-14-01461-t001:** Key estimates and assessments for CS1.

Indicator	Scale of Laboratory Trials	Scale of CS1
Input of feedstock, kg DM∙y^−1^	4533	44,800
N_los_	20%	20%
Area of premises, m^2^	73.0	373.0
The volume of premises, m^3^	230.0	1174.9
Equipment (BSF rearing), EUR	11,000	55,269
Equipment (larva processing), EUR	6050	37,772
Labour, person hours∙y^−1^	72	418

**Table 2 animals-14-01461-t002:** Key estimates and assessments for CS2.

Indicator	Scale of Laboratory Trials *	Scale of CS2
Input of feedstock, t DM∙y^−1^	4.53	9746.0
N_los_ **	23%	23%
Area of premises ***, m^2^	62.1	17,017.8
Volume of premises ***, m^3^	195.5	53,606.0
Equipment (BSF rearing) ***, EUR	9350	3,048,200
Equipment (larva processing), EUR	9439	3,077,198
Labour ***, person hours∙y^−1^	61.2	17,235

* Adjusted data from BSF trials; ** 15% more than in BSF trials, *** 15% less than in BSF trials.

**Table 3 animals-14-01461-t003:** Balance (EUR) between socio-economic benefits and costs for CS1 (BSF rearing and larva processing).

Indicator	CS1A		CS1D		CS1F	
	PV	EAA	PV	EAA	PV	EAA
**Benefits:**						
The economic value of larva meal	57,806	4101	2,890,288	205,073	2,890,288	205,073
The economic value of larva oil	20,210	1434	1,010,500	71,697	1,010,500	71,697
The economic value of frass	14,983	1063	749,137	53,153	749,137	53,153
Avoided organic waste	74,610	5294	3,730,491	264,688	3,730,491	264,688
Avoided GHG emissions from fish meal production	4553	323	227,645	16,152	227,645	16,152
Avoided adverse impact on biodiversity and ecosystems from reduction fishery	N.A.	N.A.	N.A.	N.A.	N.A.	N.A.
**Total**	**172,161**	**12,215**	**8,608,061**	**610,763**	**8,608,061**	**610,763**
**Costs:**						
Energy (electricity, heat)	63,341	4494	2,883,991	204,626	2,808,078	199,240
Water and sewage	1448	103	72,384	5136	72,384	5136
Consumables	96,698	6861	2,201,387	156,194	4,286,883	304,165
Labour cost	86,689	6151	1,455,189	103,249	1,455,189	103,249
GHG emissions from consumed energy	1240	88	62,018	4400	62,018	4400
Net economic value of substituted fishmeal and fish oil	14,180	1006	708,984	50,304	708,984	50,304
**Total**	**263,595**	**18,703**	**7,383,953**	**523,910**	**9,393,537**	**666,495**
**Balance excluding investment**	**−91,434**	**−6487**	**1,224,108**	**86,853**	**−785,476**	**−55,731**
Investment (insect rearing, larva processing)	750,469	53,248	13,380,247	949,361	6,690,123	474,681
**Balance**	**−841,903**	**−59,735**	**−12,156,139**	**−862,508**	**−7,475,599**	**−530,412**

PV—present value; EAA—equal annual annuity; CS1A, CS1D, and CS1F—situations of what-if analysis (see Section 2.3); N.A.—not assessed monetary.

**Table 4 animals-14-01461-t004:** Balance (EUR) between socio-economic benefits and costs for CS2 (YM rearing and larva processing).

Indicator	CS2A		CS2B		CS2F	
	PV	EAA	PV	EAA	PV	EAA
**Benefits:**						
The economic value of larva meal	24,202,252	1,717,209	24,202,252	1,717,209	242,022,525	17,172,093
The economic value of larva oil	5,538,692	392,984	5,538,692	392,984	55,386,917	3,929,838
The economic value of frass	7,206,047	511,287	7,206,047	511,287	72,060,469	5,112,867
Avoided organic waste	0	0	0	0	0	0
Avoided GHG emissions from fish meal production	1,372,775	97,402	1,372,775	97,402	13,727,755	974,018
Avoided adverse impact on biodiversity and ecosystems from reduction fishery	N.A.	N.A.	N.A.	N.A.	N.A.	N.A.
**Total**	**38,319,767**	**2,718,882**	**38,319,767**	**2,718,882**	**383,197,666**	**27,188,816**
**Costs:**						
Energy (electricity, heat)	10,039,493	712,327	10,039,493	712,327	97,751,017	6,935,675
Water and sewage	642,355	45,577	642,355	45,577	6,268,926	444,796
Consumables	13,027,561	924,337	6,513,781	462,169	126,844,785	8,999,949
Labour cost	12,606,735	894,479	12,606,735	894,479	66,313,603	4,705,113
GHG emissions from consumed energy	49,202	3491	49,202	3491	492,019	34,910
Net economic value of substituted fishmeal and fish oil	4,055,060	287,716	4,055,060	287,716	40,550,599	2,877,165
**Total**	**40,420,406**	**2,867,927**	**33,906,625**	**2,405,758**	**338,220,948**	**23,997,607**
**Balance excluding investment**	**−2,100,639**	**−149,046**	**4,413,141**	**313,123**	**44,976,718**	**3,191,209**
Investment (insect rearing, larva processing)	65,888,285	4,674,936	65,888,285	4,674,936	359,835,571	25,531,218
**Balance**	**−67,988,925**	**−4,823,981**	**−61,475,144**	**−4,361,813**	**−314,858,853**	**−22,340,009**

PV—present value; EAA—equal annual annuity; CS2A, CS2B, and CS2F—situations of what-if analysis (see Section 2.4); N.A.—not assessed monetary.

## Data Availability

The original contributions presented in the study are included in the article and the Appendix A, further inquiries can be directed to the corresponding author.

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
