# Peer review of "Assessing the Socio-Economic Benefits and Costs of Insect Meal as a Fishmeal Substitute in Livestock and Aquaculture"

_animals, 2024, doi:10.3390/ani14101461_

Round 1
Reviewer 1 Report
Comments and Suggestions for Authors
This paper assesses the socio-economic benefits and costs of production of insect meal to substitute fishmeal in feed, also focusing on factors that affect performance most. The study evaluates the economic value of insect-based products, waste, and greenhouse gas reduction as socio-economic benefits. Using data from laboratory trials and case studies on Black Soldier Fly and Yellow Mealworm. The research is original, expresses flow, and has no obvious grammatical errors. The references are appropriate, and the contents of tables and figures and the quality of the data are ok. Moreover, The conclusions are consistent with the evidence and arguments presented, and all main questions posed were addressed, but they need to be minor revision before formal acceptance.
1) The title of the article is too broad to suggest revise it, because of only involves Black Soldier Fly and Yellow Mealworm
2) I think the Introduction section is too long, needs to be streamlined, and some of this could be placed in the discussion section
3)The discussion section is too short to adequately explain the findings of the study.
Comments on the Quality of English LanguageThe quality of English language is ok
Author Response
Thank you for the valuable suggestions and recommendations. We have taken them into consideration and made improvements, where necessary. Cover letter with some explanations is attached.

Reviewer 2 Report
Comments and Suggestions for Authors
Review for the paper “Assessing the Socio-economic Benefits and Costs of Insect Meal as a Fishmeal Substitute in Livestock and Aquaculture” by Alberts Auzins, Ieva Leimane, Rihard Reissaar, Jostein Brobakk, Ieva Sakelaite, Mikelis Grivins, Lauma Zihare submitted to “Animals”.
Blue foods, commonly referred to as aquatic foods derived from marine or freshwater capture or aquaculture, have played a significant role in human consumption. As the global population continues to grow, along with wealth development and a shift towards a pescatarian diet, there will be an increasing global demand for blue foods in the following decades. It is estimated that, if capture fisheries were to apply technological and institutional innovations, they could achieve maximum production by 2050. As a result, the growing demand for global fish foods will be mainly met by aquaculture production. The continuous growth of aquaculture, driven by the rise in demand for blue food, will meet the rising demand for aquafeed. However, the use of forage fish from marine resources, which will reach their ecological stock boundary by 2040, is not long-term sustainable given the finite nature of this resource. Insect meals possess advantageous properties, including a favorable nutritional composition (high protein content and balanced amino acid profiles), health improvement for fed organisms, and environmental benefits associated with insect-containing diets compared with insect-free diets. These benefits include economic fish in fish out, land use, and solid phosphorus waste. The authors conducted a socio-economic study to assess the benefits and costs of insect meal as an alternative to costly fishmeal in livestock and aquaculture. They employed models to reveal the effects of different scenarios on the balance in two systems under financial conditions of Nordic countries, Lithuania, and Norway. They proposed a new approach to avoid the disadvantages associated with the use of the conversion ratio and to calculate the potential benefits and costs more accurately. The authors discovered that the production of insect-based products and associated processes resulted in a negative balance, indicating potential for further development of the industry, including production scaling and political initiatives, as well as state support of manufacturers in order to achieve a green economy.
The authors provided a detailed description of the methods, but the discussion should be improved by focusing on possible ways to solve the current problems in the industry. Additionally, some revisions are required to explain controversial issues.
Recommendations.
L 46-56. The first paragraph lacks references, which renders it speculative. It is recommended that the authors include relevant citations.
L 91-99. This text is devoted to the study object and hypotheses, and its inclusion in this position of the introduction seems inadequate. It is suggested that this paragraph be placed at the end of the "Introduction" section.
L 218. Prices are one of the main input variables in socio-economic models. The authors indicated that they assumed the real prices for the 2023/2024 season. It would be beneficial for the authors to provide information on trends in these prices, as this can illuminate the applicability of the authors’ results and reduce uncertainty regarding potential scenarios.
L 238. The authors asserted that labor markets in Lithuania and Norway are well developed, suggesting that the difference between market wage rates and shadow wage rates is negligible. This assertion raises a number of questions. What is meant by the term "well developed" in this context? Are these markets similarly "well developed" in Lithuania and Norway? Is there a shortage of professional staff for the aquaculture industry in Lithuania? It is necessary to include actual, relevant sources in order to provide a comprehensive analysis.
L 254. The authors should provide a rationale for the choice of fishmeal and its price as a reference point.
L 313. The authors should specify this position (L 388).
L 353, 569-571. The authors cite the paper by Cámara-Ruiz M. et al. (2023) for the average consumption values, yet they do not elucidate how the results obtained for Spain are applicable to Lithuania and Norway. Were the producing facilities comparable in terms of equipment and production stages? For example, in the current study, the authors assumed insect rearing and processing of larvae, whereas Cámara-Ruiz M. et al. (2023) considered the sacrifice, drying, and defatting of insects.
In this section, the authors cite ratios of 15.8% and 15.6% in the same context. It is necessary for them to clarify which ratio is accurate.
L 484-486. The authors should explain why they considered a 100% utilization rate for protein and oil.
Table 3. The authors should identify each abbreviation used in a footnote.
Tables 3 and 4. The authors should specify the period during which these results were obtained.
The authors posit that the benefits of the projects can be quantified as "avoiding fishing." However, this approach is open to question, given that fishmeal is produced either from low-cost fish or, more frequently, from fish discards/by-products.
L 724-743. This text is for the discussion section.
L 784-787. This repeats information from the introduction.
Comments on the Quality of English LanguageThe English should be revised because the text contains many grammar errors.
Author Response

(The authors gave the same response as above.)

Round 2
Reviewer 2 Report
Comments and Suggestions for Authors
No further comments.